# Activation of *Pvt1b* isoform contributes to local *Pvt1* abundance to repress *Myc* during stress

**Qiao Li[1], Christiane E. Olivero[1], Erin Floyd[1], Jane Ding[1], Emily Dangelmaier ⓘ[1], James Knight ⓘ[2], Nadya Dimitrova ⓘ[1]\***

**1** Department of Molecular, Cellular, and Developmental Biology, Yale University, New Haven, Connecticut, United States of America, **2** Yale Center for Genome Analysis, Yale University School of Medicine, New Haven, Connecticut, United States of America

\* nadya.dimitrova@yale.edu

## Abstract

Many long noncoding RNA (lncRNA) loci harbor multiple alternative isoforms. It is not known whether isoform-specific sequence elements enable distinct functions. Previous work identified two alternative transcription start site (TSS) isoforms in the *Pvt1* lncRNA locus - the constitutively expressed *Pvt1a* and the stress-induced *Pvt1b*. While the function of *Pvt1a* is not known, the p53-regulated *Pvt1b* was shown to act locally to repress the transcription of the neighboring *Myc* proto-oncogene in response to genotoxic and oncogenic stress. Here, we investigated whether *Pvt1b* contains isoform-specific repressive sequence elements. Our results revealed that *Pvt1b* contributes to but is not required for *Myc* repression. Using *in vivo* and *in vitro* models of genotoxic and oncogenic stress, we observed that *Pvt1a* compensates for *Pvt1b* loss, resulting in *Pvt1b* deficiency having a moderate effect on *Myc* regulation, stress response, and tumor suppression. Long-read sequencing exposed a diversity of stress-induced *Pvt1a* and *Pvt1b* isoforms, further arguing against a specialized role for *Pvt1b*. We propose that p53-induced increase in total *Pvt1* abundance, and not isoform-specific activation, represses *Myc* during stress.

## Author summary

While the functional domains and residues of many proteins are well-characterized, analogous understanding of the functional motifs within long noncoding RNAs (lncRNAs) is lacking. As a result, little is known about se-quence elements that enable the proposed diverse functions of lncRNAs. In this study, we tackled this question by genetically dissecting the locus of *Pvt1*, a tumor suppressive lncRNA, which has been shown to negatively regulate in *cis* the transcription of the neighboring Myc proto-oncogene in response to stress and during tumor development. Unexpectedly, we found that *Myc* repression is not dependent on specialized elements within the stress-specific *Pvt1* isoform,

**Data availability statement:** The data that support the findings of this study are publicly available from NCBI BioProject ID PRJNA1222429. All other relevant data are within the manuscript and its Supporting Information files.

**Funding:** This work was supported by NIH R37CA230580 (ND) and by Predoctoral Training Programs NIH T32GM007499 (QL) and NIH T32GM007223 (CEO). The funders had no role in study design, data collection and analysis, decision to publish, or preparation of the manuscript.

**Competing interests:** The authors have declared that no competing interests exist.

*Pvt1b*, but is caused by a stress-dependent increase in the overall abundance of locally produced *Pvt1* transcripts. Our findings indicate that changes in lncRNA abundance can play an important function in regulating gene expression. We propose that local accumulation of *cis*-regulatory lncRNAs modulates in a dose-dependent manner the transcriptional environment of target genes.

## Introduction

Mammalian genomes express tens of thousands of long noncoding RNAs (lncRNAs), which are transcripts exceeding 500 nucleotides in length and lacking protein-coding potential [1]. In recent years, a growing number of lncRNAs have been functionally characterized and implicated in key cellular processes, such as epigenetic, transcriptional, and post-transcriptional regulation [2]. For some lncRNAs, specific sequence elements have been linked to interactions with downstream effectors, such as Repeat A in *Xist* scaffolding Spen recruitment during X-chromosome inactivation [3], or an array of Pumilio response elements (PREs) in *Norad* sequestering Pum1/2 in cytoplasmic bodies [4]. For other lncRNAs, the acts of transcription initiation, elongation, or processing have been shown to be sufficient for their regulatory functions [5]. Whether or not most lncRNAs contain motifs and/or structural elements that enable their activities remains unknown. On the one hand, the lack of apparent evolutionary conservation in most lncRNAs has suggested lack of functional sequence elements. On the other hand, it has been proposed that short nucleotide motifs or conserved secondary structures may underlie lncRNA functions.

In response to stress, such as genotoxic damage or oncogenic signaling, the transcription factor p53 binds to p53 response elements (p53REs) in the promoters of target genes to induce a temporary cell cycle arrest or eliminate damaged cells through senescence and apoptosis [6]. Several lncRNAs have been shown to be direct transcriptional targets of p53 and to mediate its cellular outcomes [7]. Examples of p53-regulated lncRNAs include *lincRNA-p21,* which acts *in cis* to promote the expression of its neighbor, the G1/S checkpoint regulator Cdkn1a (also known as p21) [8], and *Neat1,* which acts *in trans* to limit oncogenic transformation [9]. Thus, lncRNAs have emerged as important modulators of the p53 stress response and tumor suppression network.

The *PVT1* (Plasmacytoma variant translocation 1) lncRNA is located approximately 50 kb downstream of the MYC proto-oncogene and is frequently co-amplified with *MYC* in various cancer types [7]. A study modeling *Myc/Pvt1* co-amplification in murine breast cancer initially proposed that increased *Pvt1* expression promotes Myc protein stability through a post-translational mechanism [10]. As MYC is a powerful driver of cellular proliferation in a dose-dependent manner, the conclusion was that *PVT1* is an oncogenic lncRNA [11]. However, the *PVT1* gene body also harbors multiple enhancers that promote *MYC* expression, suggesting that enhancer amplification rather than increased lncRNA expression may be the oncogenic driver. Efforts to dissociate RNA and DNA-based mechanisms unexpectedly revealed that the

*PVT1* lncRNA harbors tumor suppressive elements that limit *MYC* levels. While the *PVT1* locus contains oncogenic *MYC* enhancers, *PVT1* transcription was shown to serve a tumor suppressive role by curbing *MYC* transcription [12]. Consistent with this conclusion, cancer genome sequencing identified recurrent mutations encompassing the *PVT1* promoter [12].

Further supporting a tumor suppressive function, *Pvt1* was also identified as a p53 target [13,14]. The *Pvt1* locus gives rise to multiple isoforms, including constitutively expressed isoforms, initiated at exon 1a and termed *Pvt1a*, and stress induced isoforms, initiated at exon 1b downstream of a conserved p53RE and termed *Pvt1b* [14]. Numerous genome-wide p53 binding profiles across various cell types and in response to different stressors have confirmed binding of p53 to the *Pvt1b*-associated p53RE, indicating that *Pvt1b* is a canonical p53 target [15–19]. Consistent with p53 dependence, *Pvt1b* is not expressed in cells lacking p53 and becomes strongly upregulated in p53-proficient cells exposed to stress [14]. Importantly, p53-induced *Pvt1b* was found to act locally to repress the transcription of the neighboring *Myc* in response to stress and to play an important role in limiting cellular proliferation in the presence of genotoxic damage and oncogenic signaling [14]. These data pointed to the *Pvt1b* isoform as a mediator of *Myc* repression in the *Pvt1* locus.

However, the mechanism by which *Pvt1b* represses *Myc* was not clear. Beyond the first exon, the constitutively expressed *Pvt1a* and stress responsive *Pvt1b* isoforms span more than 10 alternatively spliced downstream exons over a 300 kb-long region. Isoform analysis has indicated a comparable pattern of downstream exon inclusion, suggesting that *Pvt1a* and *Pvt1b* primarily differ by their alternative transcription start site (TSS) [14]. It was proposed that *Pvt1b* represses *Myc* either through isoform-specific elements, located within exon 1b, or by augmenting the local *Pvt1* abundance, independent of isoform identity. In this study, to distinguish between these two models, we performed CRISPR/Cas9 mutagenesis screen for specialized repressive sequences in the *Pvt1b* endogenous locus, employed a *cis* repression reporter assay, and developed genetically engineered mice and cells with *Pvt1b*-specific inhibition. Our data point to increased *Pvt1* abundance, and not *Pvt1b* elements, as the mediator of *Myc* repression during stress.

## Results

### Mutagenesis screen to identify *Pvt1b*-specific functional elements

To probe the contribution of *Pvt1b* elements to *Myc* repression, we used CRISPR/Cas9 to introduce mutations and indels in *Pvt1b*-specific sequences (Fig 1A). We first targeted the 3' end of exon 1b to prevent splicing to exon 2. We designed and introduced a splice site (SS)-targeting gRNA (gSS) in the p53 restorable $p53^{LSL/LSL}$; $R26$-$CreER^{T2}$ (PR) mouse embryonic fibroblasts (MEFs) (Fig 1A and 1B). In PR MEFs, p53 expression is prevented by a transcriptional/translational termination (Stop) cassette flanked by loxP sites [20]. Treatment with Tamoxifen (Tam) leads to CreER activation and loxP-mediated excision of the Stop cassette, reverting the p53 locus to wild-type state. Concomitant exposure to the genotoxic agent Doxorubicin (Doxo) promotes p53 stabilization and activates the p53-mediated transcriptional response to stress. As a negative control, we introduced a non-targeting gRNA (gTom). As a positive control, we introduced a gRNA targeting the *Pvt1b*-associated p53RE (gRE), which was previously shown to abrogate stress-induced *Pvt1b* upregulation and *Myc* repression [14]. We confirmed comparable Tam- and Doxo-dependent activation of the canonical p53 target, *p21*, in gTom, gRE, and gSS PR MEFs (Fig 1C). We also confirmed that p53RE mutagenesis led to expected over 90% loss of *Pvt1b* in the presence of Doxo (Fig 1D) [14]. SS mutagenesis also significantly reduced spliced *Pvt1b* levels, indicating successful perturbation of *Pvt1b* splicing in gSS compared to gTom cells (Fig 1D). Interestingly, SS but not p53RE mutagenesis was accompanied by an increase in the expression of *Pvt1a*, suggesting a compensatory p53-dependent activation of *Pvt1a* in the presence of reduced *Pvt1b* (Fig 1D).

We investigated the effects of gRE and gSS on *Myc* levels in the absence and presence of genotoxic stress. Consistent with previous data, gRE-expressing MEFs failed to downregulate *Myc* in response to Doxo (Fig 1E) [14]. In contrast, stress-induced reduction of *Myc* levels was indistinguishable between gSS- and gTom-expressing MEFs (Fig 1E). We concluded that production of spliced *Pvt1b* is not required for *Myc* repression during stress.

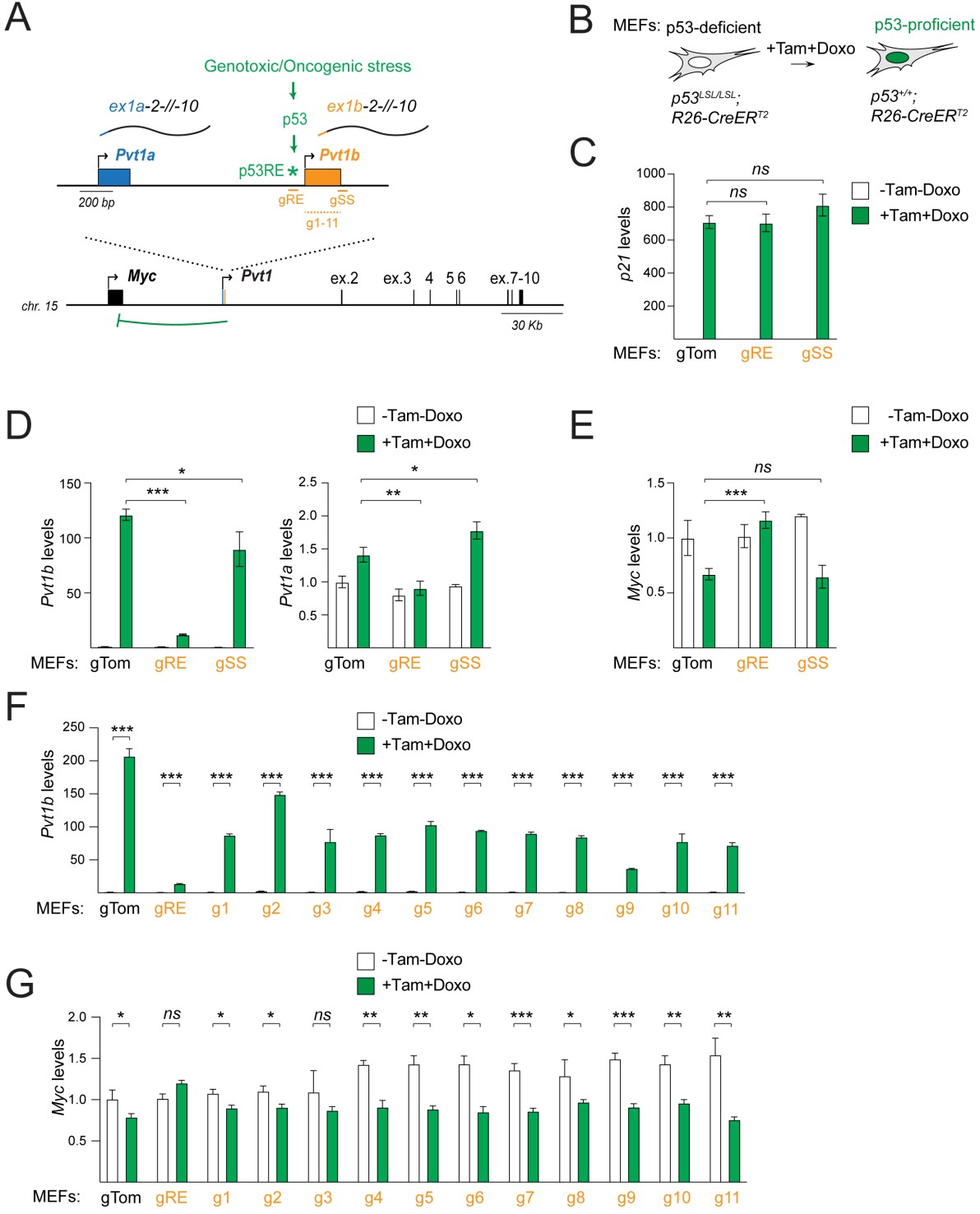

**Fig 1. CRISPR/Cas9 screen to identify *Pvt1b*-specific functional elements. A.** Schematic of murine *Myc/Pvt1* locus, illustrating *Pvt1* exons, including alternative exon 1a (blue) and exon 1b (orange), genotoxic and oncogenic stress-induced p53 activation and binding to a p53 response element (p53RE), and guide RNAs targeting p53RE (gRE), 5' splice site (gSS), and 11 sites in exon 1b (g1-11). **B.** Schematic of *p53^{LSL/LSL}; R26-CreER^{T2}* (PR) MEFs, showing Tamoxifen (Tam)-mediated p53 restoration and Doxorubicin (Doxo)-induced p53 activation. **C-E.** RT-qPCR detection of *p21* (C), *Pvt1b* and *Pvt1a* (D) and *Myc* (E) RNA levels in PR MEFs expressing gTom, gRE, or gSS in the absence and presence of Tam and Doxo treatments.

Data show mean ± SD of normalized RNA levels in n = 3 biological replicates. Paired t-test, * p < 0.05, ** p < 0.01, *** p < 0.001, *ns* not significant. **F-G.** RT-qPCR detection of *Pvt1b* (F) and *Myc* (G) RNA levels in PR MEFs expressing gTom, gRE, or g1-11 in the absence and presence of Tam and Doxo treatments. Data show mean ± SD of RNA levels normalized to untreated gTom samples in n = 3 technical replicates and was confirmed in an independent biological replicate. Unpaired t-test, * p < 0.05, ** p < 0.01, *** p < 0.001, *ns* not significant.

We next asked whether exon 1b itself harbored repressive elements. We designed 11 gRNAs (g1-g11) that spanned the length of exon 1b (Figs 1A, S1A and S1B). We introduced g1-g11 in PR MEFs and assessed the effects of exon 1b mutagenesis on stress-induced *Pvt1b* upregulation and *Myc* downregulation (Fig 1F and 1G). In contrast to p53RE mutagenesis, which inhibited *Pvt1b* expression and abrogated stress-induced *Myc* downregulation, none of the mutations in exon 1b rescued stress-induced *Myc* downregulation. Introduction of gRE, gSS, and g1-g11 had similar effects in the p53-restorable lung adenocarcinoma *K-ras*$^{G12D/+}$*; p53*$^{LSL/LSL}$*; R26-CreER*$^{T2}$ (KPR) cell line, where Tam-mediated p53 restoration activates the p53 response to oncogenic stress (S1C–S1F Fig) [19,21].

In sum, analysis of cells lacking the *Pvt1b*-associated p53RE validated prior conclusions that p53 binding and transcriptional activity in the *Pvt1* locus are required for *Myc* downregulation during stress [14]. In contrast, disruption of isoform-specific functional elements through mutagenesis of *Pvt1b* sequences failed to rescue *Myc* repression during stress.

### *Pvt1a* and *Pvt1b* isoforms show comparable activity in *cis*-repression reporter assay

Since CRISPR/Cas9 mutagenesis may have been inefficient or failed to target critical functional motifs (S1A and S1B Fig), we next probed the role of *Pvt1b* in a previously established pTetris reporter assay for RNA-mediated *cis*-regulation [22,23]. The Piggybac pTetris construct contains PGK-driven luciferase gene next to a TRE-controlled expression cassette, where RNAs of interest can be inserted (Fig 2A). Quantification of luminescence in the absence and presence of Doxycycline (Doxy)-induced RNA expression has been shown to report on RNA-mediated effects on luciferase expression *in cis* [22,23].

We generated stable PR MEF lines with empty pTetris (pTetris-EV) or pTetris constructs expressing TRE-controlled full-length *Pvt1a* and *Pvt1b*, which differ only by the inclusion of the alternative first exon (pTetris-*Pvt1a* and -*Pvt1b*, Fig 2A). We also generated a pTetris MEF line expressing *LincRNA-p21*, a control lncRNA of comparable length but implicated in *cis* activation (pTetris-*LincRNA-p21*, Fig 2A). We confirmed by quantitative PCR and copy number quantification dose-dependent induction of *Pvt1a*, *Pvt1b*, and *LincRNA-p21* in the corresponding Doxy-treated cells compared to Doxy-treated pTetris-EV or untreated controls (Fig 2B).

Endogenous *cis*-regulatory lncRNAs, including *Pvt1* isoforms, accumulate in the chromatin at their sites of transcription, where they can exert local transcriptional control of neighboring genes [24]. To determine whether transgenic *Pvt1a* and *Pvt1b* similarly accumulated at the sites of pTetris insertion, we used single molecule RNA FISH (smRNA-FISH). A probe set specific to exonic regions of *Pvt1* that are shared between *Pvt1a* and *Pvt1b* (*Pvt1e*, red) was used to recognize both endogenous and exogenous *Pvt1* transcripts, while a probe set specific to intronic regions of *Pvt1* (*Pvt1i*, green) was used to identify sites of endogenous nascent transcription (Fig 2C and 2D). In pTetris-EV cells, both in the absence and in the presence of Doxy, *Pvt1e* signals co-localized with *Pvt1i* foci, indicative of endogenous *Pvt1* loci (*Pvt1e+/Pvt1i+*, Fig 2C). The same pattern was predominantly observed in pTetris-*Pvt1a* and -*Pvt1b* cell lines in the absence of Doxy (Fig 2C and 2D). In contrast, in Doxy-treated pTetris-*Pvt1a* and -*Pvt1b* cells, we detected the endogenous loci as well as strong *Pvt1e*-positive, *Pvt1i*-negative (*Pvt1e+/Pvt1i-*) nuclear foci representing sites of pTetris insertion and transgene expression (Fig 2C and 2D). Unlike endogenous *Pvt1*, transgenic *Pvt1* transcripts disseminated throughout the nucleoplasm and cytoplasm, similarly to mRNAs and other exogenously expressed *cis*-acting lncRNAs (Fig 2C and 2D) [8]. Even though transgenic *Pvt1* transcripts were not retained in the chromatin, the intensity of *Pvt1e*-positive foci at transgenic

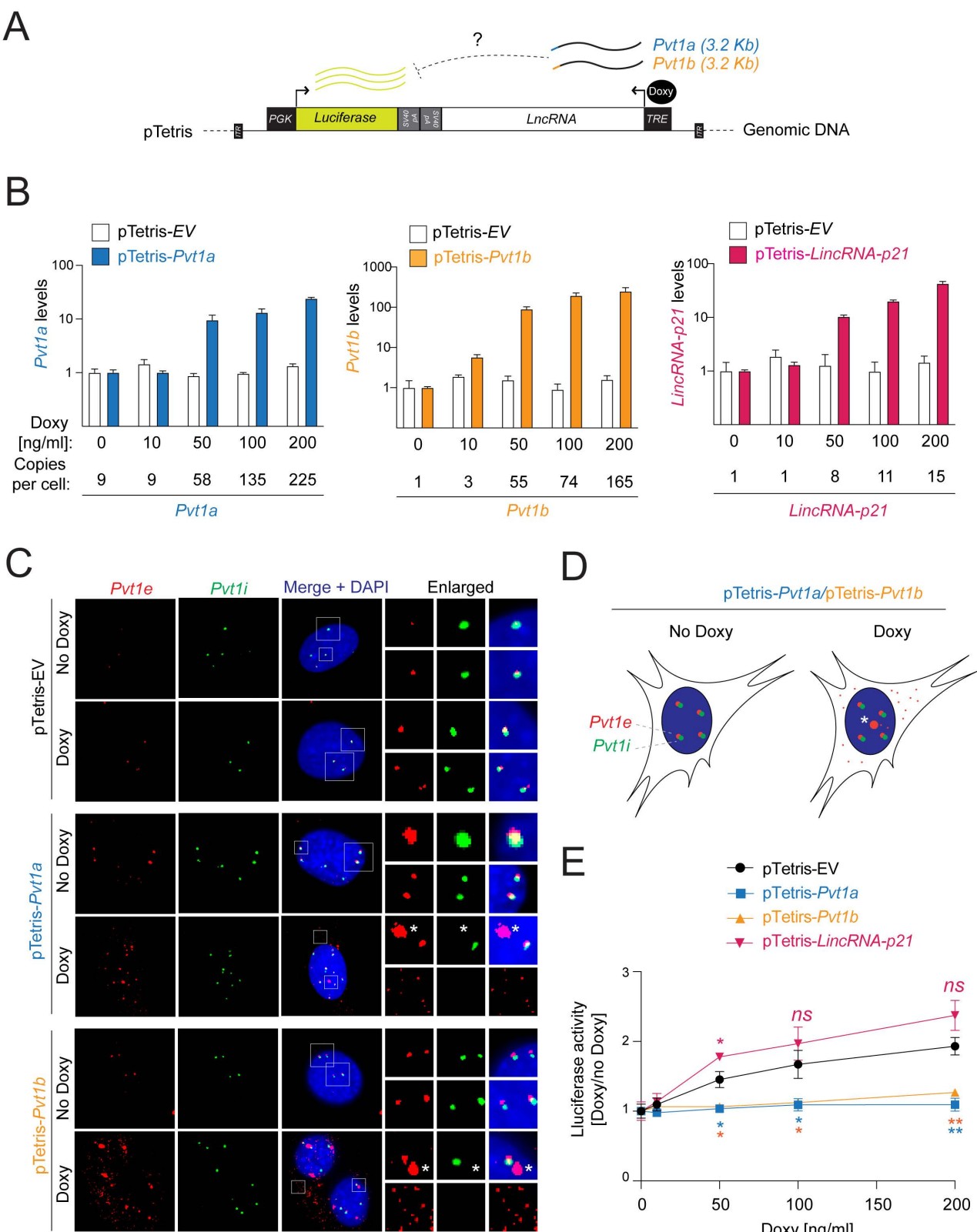

**Fig 2. Reporter assay for local *cis* repression reveals *Pvt1a* and *Pvt1b* repressive activities. A.** Schematic of genomic DNA-integrated pTetris Piggybac reporter, highlighting PGK-driven luciferase reporter downstream of a TRE-controlled lncRNA expression cassette. Luciferase activity reports

on the effect of Doxycycline (Doxy)-induced lncRNA expression on local transcription. *Pvt1a* and *Pvt1b* have comparable lengths and only differ by inclusion of alternative first exon (exon 1a, blue, or exon 1b, orange). **B.** Bargraph showing normalized levels of indicated lncRNAs in PR MEFs expressing pTetris empty vector (EV) or indicated lncRNAs and treated with indicated Doxy concentration. Numbers represent copy number of indicated lncRNAs in each sample. **C, D.** smRNA-FISH detection of endogenous and exogenous mature *Pvt1* transcripts with *Pvt1* exonic probes (*Pvt1e*, Q570, red) and endogenous nascent *Pvt1* transcripts with *Pvt1* intronic probes (*Pvt1i*, Q670, green) in indicated pTetris-EV, -*Pvt1a*, and -*Pvt1b* PR MEFs fixed in the absence or presence of 24h of 100 ng/ml Doxy treatment. DAPI, DNA. Enlarged images (C) and schematic (D) highlight endogenous *Pvt1* loci (*Pvt1e +/ Pvt1i+*), sites of pTetris insertion (*Pvt1e +/Pvt1i-*, white star), and cytoplasmic dissemination of exogenously expressed *Pvt1a* and *Pvt1b* (*Pvt1e +/Pvt1i-*). **E.** Graph showing Doxy-dependent changes in luciferase activity from luciferase assay performed in pTetris-EV, -*Pvt1a*, -*Pvt1b*, and -*LincRNA-p21* PR MEFs in the presence of 24h of indicated Doxy treatment. Data show mean±SD of n=3 biological replicates. Paired t-test, * p<0.05, ** p<0.01, *ns* not significant.

*Pvt1* transcription sites (*Pvt1e +/Pvt1i-* foci) equaled to or exceeded the intensity of *Pvt1e* signals at endogenous *Pvt1* loci (*Pvt1e +/Pvt1i+* foci) (Fig 2C). We concluded that transgenic *Pvt1a* and *Pvt1b* transcripts accumulate in the chromatin at the pTetris insertion sites at levels comparable to endogenous *Pvt1* loci.

Having validated the reporter system, we examined the roles of transgenic *Pvt1a* and *Pvt1b* in *cis*-regulation. We performed luciferase reporter assay in pTetris-EV, -*Pvt1a*, -*Pvt1b*, and -*LincRNA-p21* cells treated with increasing Doxy concentrations and observed a dose-dependent increase in luciferase activity in control EV and *LincRNA-p21*-expressing cells, establishing a baseline for this assay, which aligned with previous reports (Fig 2E) [23]. In contrast, luciferase activity was significantly reduced in both Doxy-treated pTetris-*Pvt1a* and -*Pvt1b* cells compared to Doxy-treated control pTetris-EV and -*LincRNA-p21* cells, supporting roles in *cis*-repression (Fig 2E). Notably, *Pvt1a* and *Pvt1b* reduced luciferase activity to a similar extent (Fig 2E), suggesting absence of *Pvt1b*-specific repressive elements.

### Development of *Pvt1b*-specific loss-of-function mouse model

As reporter constructs may not accurately recapitulate physiological settings, we next sought to develop a model with specific inhibition of endogenous *Pvt1b*. A prior study has reported that insertion of the 49-nucleotide synthetic polyadenylation signal (PAS) more than 1kb downstream of the TSS of a lncRNA frequently leads to inefficient termination [25]. Therefore, we reasoned that PAS insertion in exon 1b may cause premature termination of *Pvt1b* without affecting *Pvt1a* transcription. We performed CRISPR/Cas9-mediated gene editing in murine blastocysts to insert PAS (*P*) in exon 1b in the endogenous *Pvt1* locus (Fig 3A). Germline transmission of the *P* allele was confirmed by genotyping (Fig 3B). Heterozygous crosses revealed that homozygous mutant mice (*Pvt1b^{P/P}*) were born at Mendelian ratio, were fertile, and did not display any apparent developmental abnormalities (Fig 3C and 3D). RNA analysis confirmed over 90% loss of *Pvt1b* expression in RNA isolated from spleen and thymus of *Pvt1b^{P/P}* compared to *Pvt1b^{+/+}* animals (Fig 3E). Importantly, there was no change in *Pvt1a* expression levels in *Pvt1b* mutant mice (Fig 3F). We concluded that *Pvt1b^{P/P}* represents a *Pvt1b*-specific loss-of-function model.

### *Pvt1b* contributes to *Myc* repression and senescence in response to genotoxic stress

To determine whether *Pvt1b* contributes to *Myc* regulation during stress, we isolated MEFs from E13.5 littermate *Pvt1b^{+/+}* and *Pvt1b^{P/P}* embryos from heterozygous crosses and performed RNA analysis at 24 hours post mock or Doxo treatment. We confirmed a significant upregulation of *Pvt1b* in Doxo-treated *Pvt1b^{+/+}* MEFs compared to untreated cells and estimated that in *Pvt1b^{+/+}* MEFs *Pvt1b* constitutes approximately 15% of total *Pvt1* in the presence of stress (Fig 4A and 4B). We also established that PAS insertion in exon 1b effectively reduced *Pvt1b* levels over 95% in *Pvt1b^{P/P}* MEFs compared to wild-type littermate controls (Fig 4A). The effect on mature *Pvt1* levels was blunted by a compensatory activation of *Pvt1a*, which offset the loss of *Pvt1b,* resulting in a limited effect on total processed *Pvt1* (Fig 4B and 4C). There was, however, an approximately 15% decrease in nascent *Pvt1* levels in Doxo-treated *Pvt1b^{P/P}* MEFs compared to Doxo-treated wild-type controls, consistent with the fraction of total *Pvt1* represented by *Pvt1b* (Fig 4B).

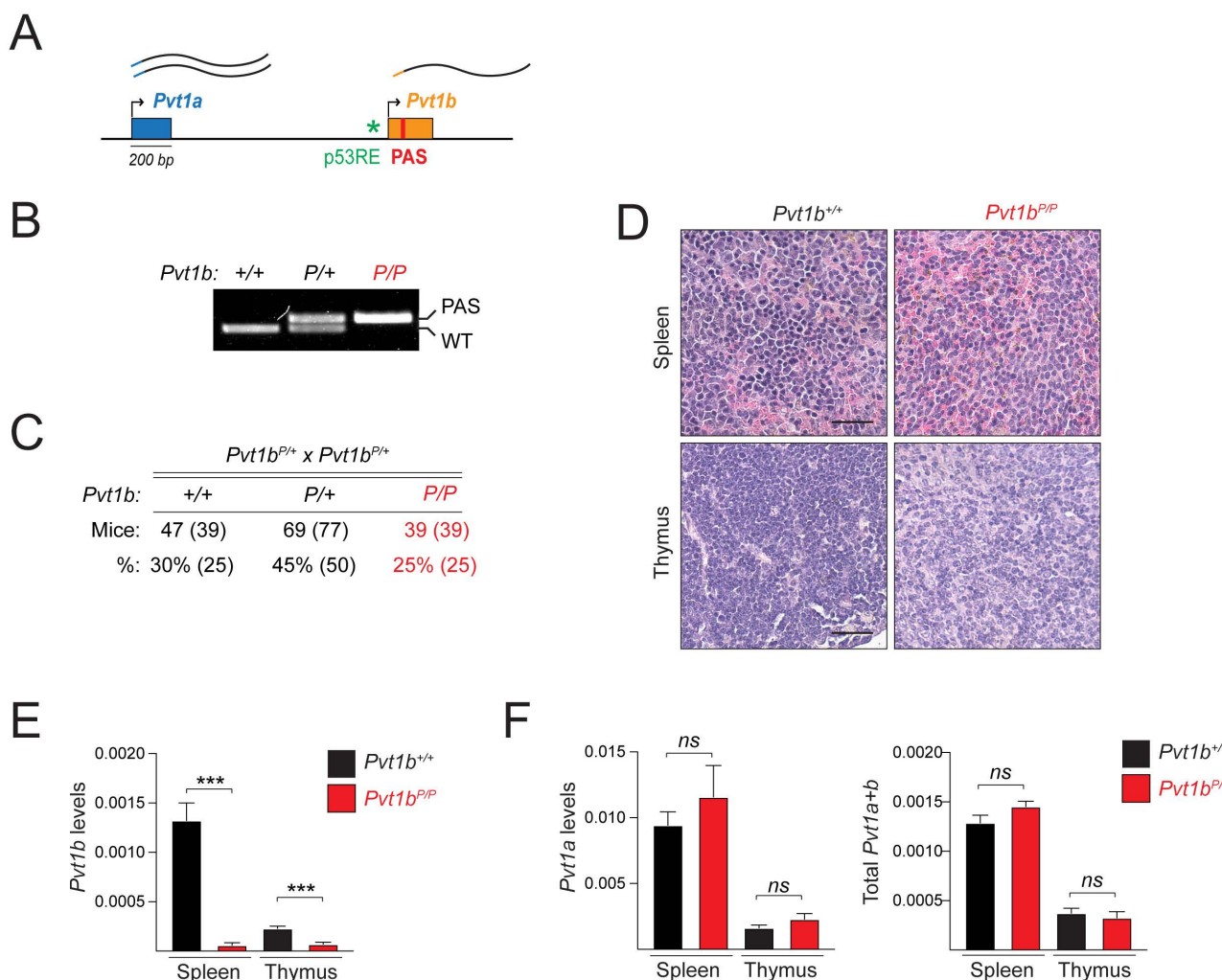

**Fig 3. Development of *Pvt1b*-specific loss-of-function mouse model. A.** Schematic of approach for the development of *Pvt1b*-specific loss-of-function mouse model by inserting early polyadenylation signal (PAS, P) in exon 1b. **B.** PCR genotyping detecting WT and PAS alleles in *Pvt1b* wild-type (*Pvt1b*<sup>+/+</sup>), heterozygous (*Pvt1b*<sup>P/+</sup>), and homozygous (*Pvt1b*<sup>P/P</sup>) mice. **C.** Mendelian distribution of indicated progeny from heterozygous crosses. **D.** H&E staining of spleen and thymus from *Pvt1b*<sup>+/+</sup> and *Pvt1b*<sup>P/P</sup> mice, showing lack of overt phenotypes. **E, F.** RT-qPCR detection of relative levels of *Pvt1b* (E), *Pvt1a* and total *Pvt1* (F) in spleen and thymus from *Pvt1b*<sup>+/+</sup> and *Pvt1b*<sup>P/P</sup> mice.

Next, we examined the impact of *Pvt1b* loss on *Myc* expression and observed a moderate but highly reproducible rescue of stress-induced *Myc* downregulation ([Fig 4D]). Whereas treatment with Doxo in *Pvt1b*<sup>+/+</sup> MEFs led to a 58±14% reduction of *Myc* RNA levels compared to mock treated cells, the reduction in *Pvt1b*<sup>P/P</sup> MEFs was only 38±18% (p<0.001, [Fig 4D]). Similarly, we observed that while in *Pvt1b*<sup>+/+</sup> MEFs Myc protein levels were reduced by 71±4% in the presence of Doxo, the reduction was 59±4% in *Pvt1b*<sup>P/P</sup> MEFs (p=0.0137, [Fig 4E] and [4F]). This difference was not due to diminished activation of the p53-mediated stress response as *p21* RNA and protein were upregulated to a similar extent in wild-type and mutant MEFs ([Fig 4F] and [4G]).

We also examined how *Pvt1b* deficiency affected the cellular response to stress. Quantification of senescence-associated β-galactosidase-positive *Pvt1b*<sup>+/+</sup> and *Pvt1b*<sup>P/P</sup> cells following a chronic exposure to low levels of Doxo revealed a significant reduction in the fraction of senescent cells in mutant compared to wild-type cells ([Fig 4F]). We also observed increased levels of the mitotic marker phosphorylated histone H3 (pHH3), and a significant reduction in the senescence

PLOS Genetics

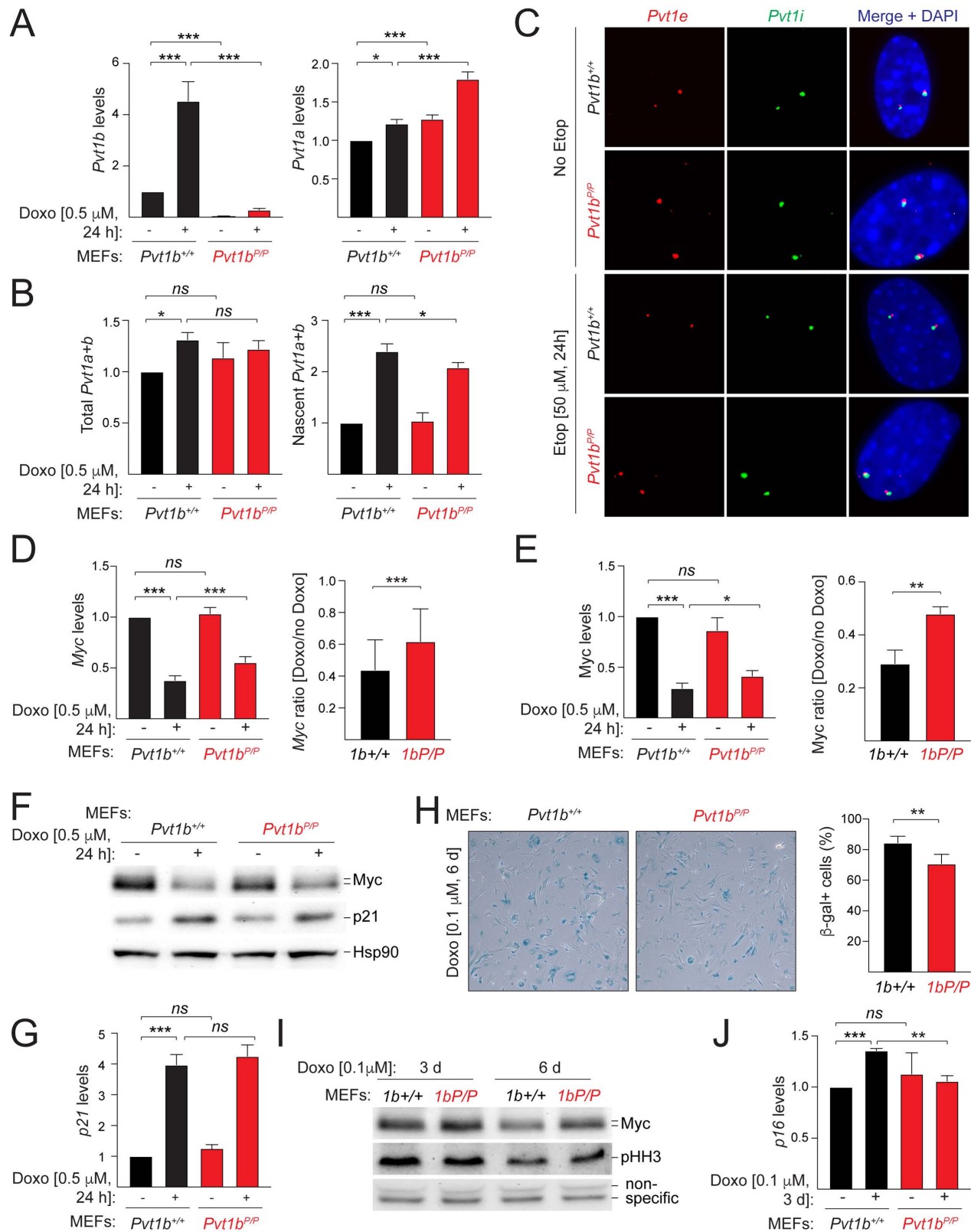

**Fig 4. *Pvt1b* partially contributes to genotoxic stress response. A, B.** RT-qPCR detection of normalized levels of *Pvt1b* and *Pvt1a* (A) and mature and nascent total *Pvt1* (B) in *Pvt1b⁺/⁺* and *Pvt1bᴾ/ᴾ* MEFs harvested in the absence or presence of 24 h with 0.5 μM Doxo. **C.** smRNA-FISH detection

of *Pvt1* with *Pvt1* exonic (*Pvt1e*, red) and intronic probes (*Pvt1i*, green) in *Pvt1b^+/+* and *Pvt1b^P/P* MEFs treated with mock or 50 μM Etoposide (Etop) for 24 h. DAPI, DNA. **D.** *Left,* RT-qPCR detection of normalized levels of *Myc* RNA levels in cells in (A); *Right,* Quantification of Doxo-induced change in *Myc* RNA levels in *Pvt1b^+/+* and *Pvt1b^P/P* MEFs. **E.** *Left,* Quantification of normalized Myc protein levels in cells in (A); *Right,* Quantification of Doxo-induced change in Myc protein levels in *Pvt1b^+/+* and *Pvt1b^P/P* MEFs. **F.** Representative immunoblot showing Myc and p21 protein levels in cells in (A). Hsp90 as a loading control. **G.** RT-qPCR detection of normalized levels of *p21* in cells in (A). **H.** Representative brightfield images and quantification of senescence-associatedβ-galactosidase-positive cells in *Pvt1b^+/+* and *Pvt1b^P/P* MEFs treated for 1 week with 0.1 μM Doxo. **I.** Immunoblot showing Myc and pHH3 protein levels in *Pvt1b^+/+* and *Pvt1b^P/P* MEFs, treated as indicated. **J.** RT-qPCR detection of normalized levels of *p16* in cells in (I). Data show mean ± SD of n ≥ 3 biological replicates. Paired t-test, * p < 0.05, ** p < 0.01, *** p < 0.001, *ns* not significant.

marker *p16* RNA levels in Doxo-treated *Pvt1b^P/P* compared to *Pvt1b^+/+* cells (Fig 4I and 4J). These findings are consistent with prior work indicating that *Pvt1b* promotes cellular senescence [19].

### *Pvt1b* limits tumor growth in lung cancer model

Previous studies reported that mutagenesis of the *Pvt1b*-associated p53RE in a *K-ras^G12D* mouse model of lung cancer leads to a significant increase in tumor burden, pointing to a tumor suppressive function [14]. To determine the contribution of *Pvt1b* to tumor suppression *in vivo*, we crossed *Pvt1b^P/P* mice to the previously established *K-ras^LA1* lung cancer mouse model (LA1). In LA1 mice, spontaneous recombination of a latent *K-ras^G12D* allele in lung epithelial cells leads to the synchronous development of 20–40 lung lesions in 100% of the animals [26]. We generated a cohort of *LA1; Pvt1b^+/+* (n = 23) and *LA1; Pvt1b^P/P* (n = 28) littermate mice and performed histopathological analysis of tumor grade and burden in hematoxylin and eosin (H&E)-stained lung sections at 5 months of age. We observed multiple lung lesions in both *LA1; Pvt1b^+/+* and *LA1; Pvt1b^P/P* mice, with approximately 70–75% of lesions categorized as AAH and grade 1 and approximately 20–25% representing grade 2–3 (Fig 5A and 5B). There was no apparent difference in grade between *Pvt1b* wild-type and mutant mice, consistent with previous findings (Fig 5B) [14]. Tumor burden quantification, however, revealed a two-fold increase in the fraction of lung tissue occupied by tumors in LA1; *Pvt1b^P/P* (4.4 ± 1.8%) compared to LA1; *Pvt1b^+/+* (2.4 ± 1.4%) mice (p = 0.0038, Fig 5A and 5C). The increase in tumor burden correlated with increased intensity of nuclear Myc staining in tumors from LA1; *Pvt1b^P/P* compared to LA1; *Pvt1b^+/+* animals (Fig 5D and 5E), consistent with *Pvt1b* loss leading to increased Myc expression. These findings demonstrated that *Pvt1b* limits lung cancer growth and contributes to tumor suppression *in vivo* through Myc regulation.

### Long-read sequencing reveals a diversity of stress-induced *Pvt1* isoforms

Lastly, we considered whether the compensatory *Pvt1a* upregulation, observed in gSS and *Pvt1b^P/P* mutants, might account for the moderate effects of *Pvt1b*-deficiency on *Myc* levels, cellular senescence, and tumor growth. To characterize the isoforms produced from the *Pvt1* locus in the presence of stress, we performed long-read sequencing in the KPR lung adenocarcinoma cell line in the absence and presence of Tam-mediated p53 restoration and response to oncogenic stress (Fig 6A). We selected this cell line because *Pvt1* isoforms are transcribed in high copy number due to amplification of the *Myc*/*Pvt1* locus in extrachromosomal DNA (ecDNA) [14]. We reasoned that the high abundance of *Pvt1* transcripts will support a better sequencing coverage for characterizing splice variants. Importantly, it has been shown that p53 restoration in Tam-treated KPR cells leads to *Pvt1b* upregulation and *Myc* repression (Fig 6B), indicating that the p53-*Pvt1b*-*Myc* regulatory axis is not perturbed by the ecDNA context [14].

Analysis of long reads in mock and Tam-treated KPR cells revealed 18 *Pvt1* isoforms, which were detectable at ≥ 5 reads in at least one condition (S1 Table). All 18 isoforms initiated at exon 1a or exon 1b, while inclusion of downstream exons was variable (Fig 6C). We classified *Pvt1* isoforms into three groups, based on changes in their abundance in response to Tam-induced oncogenic stress (Fig 6C and 6D). The stress-independent *Pvt1* isoforms in Group I represented constitutively expressed *Pvt1a* transcripts as they initiated at exon 1a and accounted for two thirds of total *Pvt1* reads.

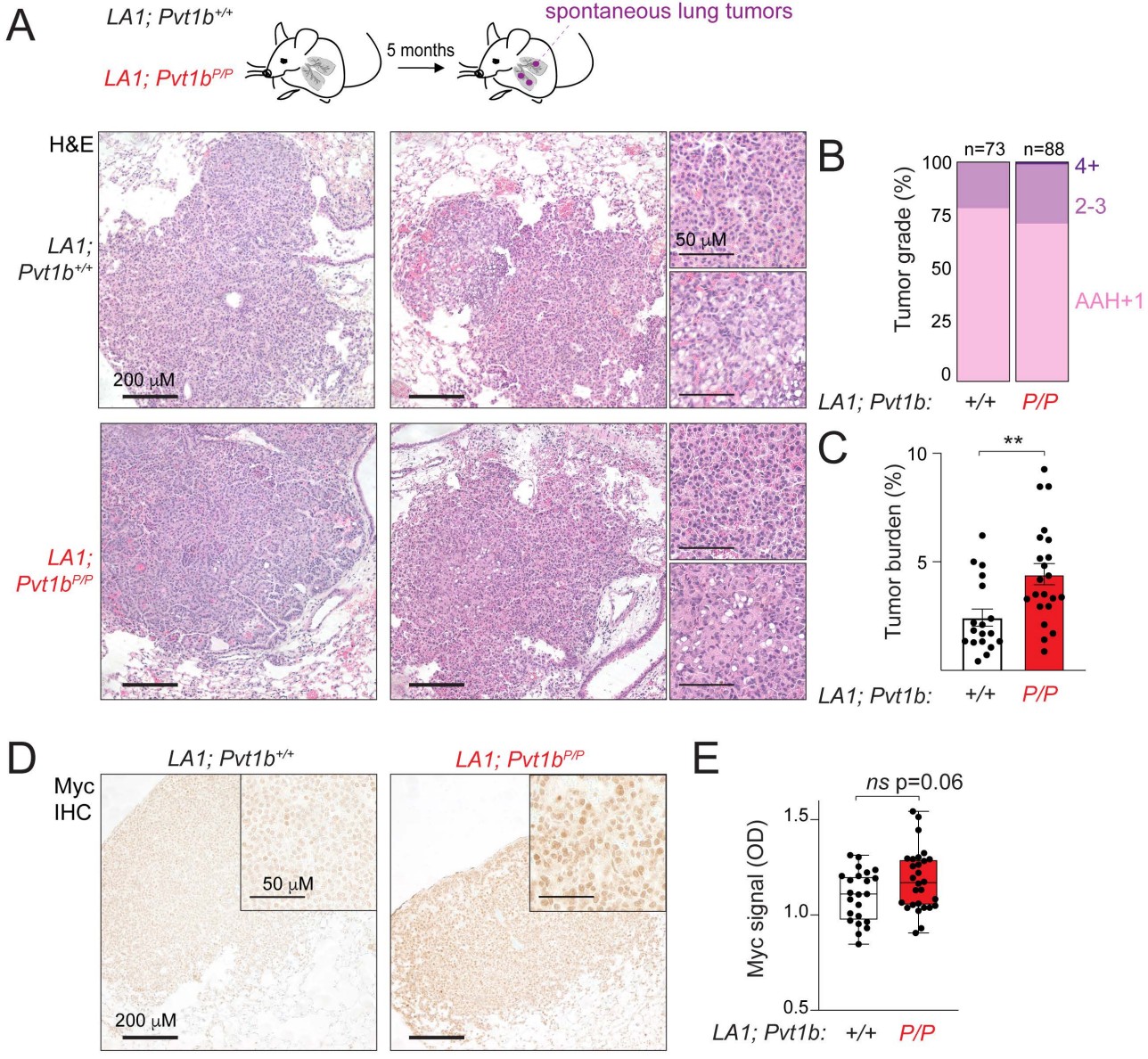

**Fig 5. *Pvt1b* limits lung cancer growth. A.** *Top* Schematic of spontaneous lung tumor development in *LA1; Pvt1b⁺/⁺* (n = 23) and *LA1; Pvt1bᴾ/ᴾ* (n = 28) mice at 5 months. *Bottom* Representative H&E images in indicated mice. Enlarged images highlight grade 1 and grade 2 tumors. **B.** Quantification of the fraction of tumors assigned to AAH-1, 2-3, or 4 + grade categories in mice from (A). Numbers above each bar indicate total number of tumors scored in each condition. **C.** Quantification of tumor burden in mice from (A). **D, E.** Representative images (D) and quantification (E) of immunohistochemistry (IHC) detection of Myc levels in tumor-bearing lungs from (A). OD (optical density) (n > 20 tumors, p = 0.06).

Interestingly, the stress-induced *Pvt1* isoforms in Group III included both exon 1a- and exon 1b-initiated isoforms. The total reads of the two stress-induced, exon 1a-containing isoforms accounted for approximately one third of stress-induced *Pvt1* transcripts, suggesting that *Pvt1a* isoforms also contributed to the local transcriptional response to stress. Finally, a previously unappreciated Group III comprised of exon 1a-initiated isoforms that were repressed by oncogenic stress. This analysis revealed an unexpected complexity of *Pvt1* isoform identity and stress-responsive expression pattern. In particular, our finding that both *Pvt1a* and *Pvt1b* isoforms are upregulated in the presence of stress is consistent with a model

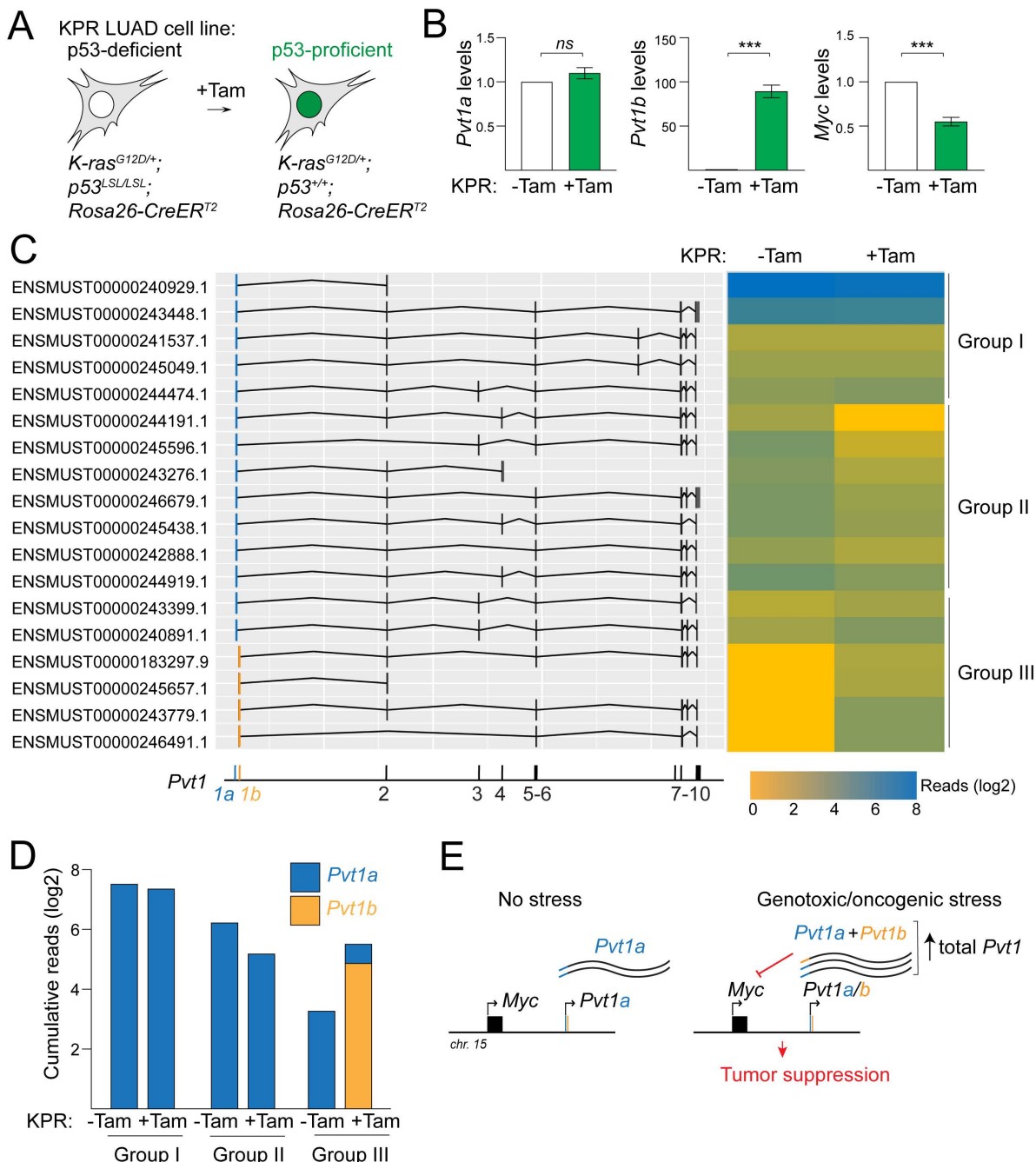

**Fig 6. Contribution of *Pvt1a* and *Pvt1b* isoforms to oncogenic stress response. A.** Schematic of *Kras^{G12D/+}; p53^{LSL/LSL}; Rosa26-CreER^{T2}* (KPR) lung adenocarcinoma (LUAD) cell line, showing Tamoxifen (Tam)-mediated p53 restoration and activation in the presence of oncogenic stress. **B.** RT-qPCR detection of normalized *Pvt1a*, *Pvt1b*, and *Myc* RNA levels in RNA isolated from mock or Tam-treated KPR cells. **C.** Schematic and heatmap of isoform abundance of *Pvt1* transcripts detected in mock and Tam-treated KPR cells by long read sequencing. Schematic highlights inclusion of exon 1a (blue) or exon 1b (orange). Heatmap represents read counts detected in the absence (-Tam) and in the presence of Tam (+Tam). Isoforms are clustered by their responsiveness to Tam: Group I (no change), Group II (repressed) and Group III (induced). **D.** Cumulative reads of exon 1a- and exon 1b-containing *Pvt1* transcripts in indicated samples and groups. **E.** Proposed model, highlighting the role of combined *Pvt1a* and *Pvt1b* abundance as the mediator of local *Myc* repression and tumor suppression.

where the two isoforms play redundant functions and where *Myc* repression is mediated by an overall increase in total stress-induced *Pvt1* transcripts and not specialized isoforms (Fig 6F).

## Discussion

In this study, we examined whether stress-induced *Pvt1b* mediates *Myc* repression through isoform-specific sequence elements. We envisioned that such elements may contain binding motifs or secondary structures that may serve as scaffolds for the local recruitment of transcriptional repressors [5]. Data from an array of complementary approaches eliminated the possibility that exon 1b contains specialized repressive elements. Specifically, analyses of *Pvt1b*-deficient mice and cells determined that *Pvt1b* contributes to but is not required for stress-induced *Myc* downregulation. Furthermore, mutagenesis screen and luciferase reporter assay indicated that *Pvt1b* does not harbor isoform-specific repressive elements. Instead, the compensatory upregulation of *Pvt1a,* observed in gSS and *Pvt1b^{PIP}* mutants, and the detection of stress-induced *Pvt1a* and *Pvt1b* by long read sequencing consistently pointed to *Pvt1a* and *Pvt1b* having additive functions in mediating *Myc* repression. We conclude that an increase in overall *Pvt1* abundance, and not isoform-specific *Pvt1b* activation, is responsible for *Myc* repression.

It remains to be determined how increased *Pvt1* abundance leads to local *Myc* repression. One possibility is that increased *Pvt1* transcription over *Pvt1* intragenic *Myc* enhancers may limit their engagement with the *Myc* promoter, as previously proposed [12], although analysis of gRE mutants did not support this model [14]. An alternative possibility is that increased local concentration of *Pvt1* molecules in response to stress may influence the transcriptional dynamics of the *Myc* promoter by precipitating the dissolution of transcriptional condensates at the *Myc* promoter, as postulated in a recently proposed RNA-mediated negative feedback model [27].

While our work led to the conclusion that *Pvt1b*-specific elements are not essential for *Myc* repression, we found that *Pvt1b* contributes to the overall local *Pvt1* abundance and regulates *Myc* in response to genotoxic and oncogenic stress. Indeed, in absence of *Pvt1b*, we observed a moderate but significant decrease in cellular senescence in response to chronic genotoxic stress and a two-fold significant increase in tumor burden in a mutant K-ras-driven mouse model of lung cancer. These results support prior conclusions that p53-mediated binding and transcriptional activity in the *Pvt1* locus is an important stress response and tumor suppressor mechanism [14,19]. One prediction is that combined loss of *Pvt1a* and *Pvt1b* expression will phenocopy deletion of the *Pvt1b*-associated p53RE, which was previously shown to disengage stress-induced p53 activation from *Myc* downregulation and cell growth inhibition [14].

## Materials and methods

### Ethics statement

All animal procedures were conducted with the approval of the Yale University Institutional Animal Care and Use Committee (IACUC).

### Mouse strains

*Pvt1b* PAS *(P)* mice were generated using CRISPR/Cas9-mediated engineering in C57BL/6J blastocysts at Jackson Laboratory. Briefly, embryos were electroporated with Cas9, a guide RNA targeting exon 1b of *Pvt1* and PAS homology directed (HDR) templates, described in S2 Table. Founders were crossed to wild-type C57BL/6J mice. Germline transmission was identified by PCR genotyping using primers in S1 Table. Correct alleles were confirmed by Sanger sequencing. Previously described *K-ras^{LA1}* mice [26] were generously provided by Dr. Jonathan Kurie (MD Anderson Cancer Center).

### Cell lines and drug treatments

All cells were maintained at 37°C in a humidified incubator with 5% $CO_2$. Primary MEFs were isolated from E13.5 embryos, resulting from timed matings between *Pvt1b^{P/+}* heterozygous mice. All experiments with primary MEFs were performed between passages 2 and 8. Primary MEFs were maintained in DMEM (Gibco) supplemented with 15% fetal bovine serum (FBS), 50 U/mL penicillin-streptomycin, 2 mM L-glutamine, 0.1 mM non-essential amino acids,

and 0.055 mM β-mercaptoethanol. Previously described p53-restorable PR *(p53^{LSL/LSL}; Rosa26-CreER^{T2})* MEFs, KPR *(Kras^{G12D/+}; p53^{LSL/LSL}; Rosa26-CreER^{T2})* murine lung adenocarcinoma cells and 293 cells were maintained in DMEM (Gibco) supplemented with 10% fetal bovine serum (FBS), 50 U/mL penicillin-streptomycin, 2 mM L-glutamine, and 0.1 mM non-essential amino acids.

CRISPR/Cas9 mutagenesis in PR and KPR cells was performed with gRNAs, listed in S2 Table, cloned downstream of the U6 promoter in BRD001 lentiviral vector (a gift from the Broad Institute, MIT) that co-expresses spCas9 and an IRES-driven puromycin resistance gene. Lentivirus was produced in 293 cells by co-transfecting BRD1 constructs with pCMV-dR8.2 dvpr (Addgene #8455) and pCMV-VSV-G (Addgene #8454) viral packaging constructs. Virus-containing supernatants supplemented with 4 µg/ml polybrene (Millipore Sigma) were used to infect PR and KPR cells by 3 consecutive lentiviral infections, delivered at 24 hr-intervals, followed by selection with 2 µg/ml and 5 µg/ml puromycin (Sigma-Aldrich), respectively.

Commercially synthesized (IDT) or genomic DNA amplified (PrimeSTAR HS, Takara) lncRNAs were cloned in NotI and ClaI sites downstream of the tetracycline-responsive promoter element (TRE) in the pTetris-cargo-Stop Piggybac vector (a gift from R. Young, MIT). Constructs were verified by restriction digestion and Sanger sequencing. Reverse tetracycline-controlled transactivator (rtTA) and pTetris Piggybac constructs were consecutively co-introduced with Piggybac Transposase in PR MEFs using the Attractive Fast-Forward Transfection protocol (Qiagen), followed by selection with 300 µg/ml G418 for 1 week or 2 µg/ml Puromycin for 4 days, respectively.

Cells were treated with 0–200 ng/ml Doxycycline, 0.5-1 µM Tamoxifen, 0.1-0.5 µM Doxorubicin, and 50 µM Etoposide for indicated time.

## RNA isolation, quantitative RT-PCR, and copy number calculations

Total RNA was isolated with RNeasy Mini Kit (Qiagen). On-column DNAse digestion was performed with RNAse-free DNase Kit (Qiagen) for analysis of nascent transcripts. 1 µg of total RNA was reverse transcribed using High-Capacity cDNA Reverse Transcription Kit (Applied Biosystems). SYBR Green PCR Master Mix (Applied Biosystems) was used for quantitative PCR in triplicate reactions with primers listed in S2 Table. Relative RNA expression levels were calculated using the ddCt method compared to Gapdh and normalized to control samples. To quantify lncRNA copy number per cell, a standard curve of the relationship between cDNA copy number and CT value was generated by quantitative PCR for each primer set and used to determine the copy number of lncRNA in RNA isolated from a known number of cells.

## Single-molecule FISH (smRNA-FISH)

smRNA-FISH was performed according to the manufacturer recommendations with previously described Quasar570 (Q570)-conjugated *Pvt1e* and Quasar670 (Q670)-conjugated *Pvt1i* probe sets (Stellaris, Biosearch Technologies) (S2 Table) [14]. Briefly, cells were plated on coverslips and treated with 50 µM Etoposide for 24 hr prior to fixation for 10 min in 4% methanol-free formaldehyde (Fisher Scientific) at RT, followed by PBS washes. Cells were dehydrated overnight at 4°C in 70% EtOH (diluted in DEPC-H$_2$O) and stored in 70% EtOH for up to a week at 4°C. Coverslips were transferred to a hybridization chamber and equilibrated for 5 min in Wash Buffer A (Stellaris, LGC Biosciences) prepared with formamide (Sigma-Aldrich) according to manufacturer's instructions. Cells were incubated overnight at 30°C with the indicated probes diluted 1:50 in Hybridization solution (Stellaris, Biosearch Technologies) prepared with formamide according to manufacturer's instructions. The next day, cells were washed 2 times for 30 min at 30°C in Wash Buffer A, incubated in Wash Buffer B (Stellaris, Biosearch Technologies) for 5 min at RT, and mounted in antifade reagent (Vectashield Mounting medium with DAPI, Vector Laboratories). Images were captured using an Axio Imager 2 microscope system (Zeiss) with a PlanApo 63x 1.4 oil DIC objective lens (Zeiss). All images were edited using Adobe Photoshop to highlight smRNA-FISH-specific signals.

## Immunoblotting

Cells were collected, counted, and lysed in 2xLaemmli buffer (100mM Tris-HCl pH6.8, 200mM DTT, 3% SDS, 20% glycerol) at $1\times10^4$ cells/µl. Samples were heated at 95°C for 7min and passed through an insulin syringe. Protein from $1\times10^5$ cells was separated on 10% SDS- polyacrylamide gels and transferred to nitrocellulose membranes (Bio-Rad). After blocking (5% milk, PBST), membranes were incubated overnight at 4°C in primary antibodies: c-Myc (1:1,000, clone Y69, ab32072, Abcam), p21 (1:200, clone F-5, sc-6246, Santa Cruz), pHH3 (1:1000, 9701S, Cell Signaling Technology), and anti-Hsp90 (1:5,000, 4877S, Cell Signaling Technology), diluted in 5% milk/PBST, washed 3 times in PBST, then incubated for 1hr at RT in secondary antibody, diluted in 5% milk/PBST (1:10,000, Jackson ImmunoResearch). After three washes in PBST, protein bands were visualized using Amersham ECL Prime Western Blotting Detection Reagent (GE Healthcare). Quantification of Myc and Hsp90 protein levels was performed using Measure Tool in ImageJ and Myc levels were normalized relative to Hsp90 levels.

## Cellular assays

Luciferase reporter assay was performed with $2–3\times10^4$ cells per well seeded in a 24-well plate and treated with 0–200ng/ml Doxycycline for 24hr. Luciferase activity was measured using the Luciferase Assay (Promega) according to manufacturer instructions in technical triplicates in at least four biological replicates. Luciferase activity in Doxy-treated was normalized to untreated cells. Senescence-associated β-galactosidase activity assay was performed at pH 5.5 as previously described in [28] with cells grown in regular media supplemented with 0.1 µM Doxorubicin for 1 week.

## Tissue analysis

Spleen and thymus dissected from adult ≥1 month old *Pvt1b+/+* and *Pvt1bP/P* mice and 4% formaldehyde-inflated tumor-bearing lungs from 5-month old *LA1; Pvt1b+/+* and *LA1; Pvt1bP/P* mice were fixed in 4% formaldehyde for 24 hrs prior to dehydration in 70% ethanol. Fixed tissues were embedded in paraffin, sectioned, and stained with hematoxylin and eosin (H&E) by Yale Pathology Tissue Services (YPTS). Tumor burden, quantified as the fraction of total lung area occupied by tumors, was determined in ImageJ. Tumor grade was scored using previously described criteria [29,30].

## Immunohistochemistry

Immunohistochemistry staining was carried out on paraffin-embedded tissue sections using Vectastain Elite ABC Peroxidase kit (Vector Laboratories, PK6101). Antigen retrieval was carried out by heating in a steamer with 10mM Citrate buffer (pH 6.0) for 30min at 95°C. Endogenous peroxidase activity was blocked with Dual Endogenous Enzyme Block (Dako, S2003), followed by Avidin/Biotin block (Vector Laboratories, SP2001), and CAS-Block (Invitrogen, 008120). Tissues were incubated with c-Myc antibody (1:100, clone Y69, ab32072, Abcam) at 4°C overnight. The signal was visualized with DAB (Vector Labs). Myc signal was quantified as optical density (OD) using the Measure Tool in Image J.

## Long read sequencing

Total RNA was isolated from KPR cells at 24 hr post mock or 1 µM Tamoxifen treatment.

PolyA selection and cDNA library preparation were conducted using the PacBio RS technology at Yale Center Genome Analysis (YCGA). Raw sequencing reads were assessed using FastQC (https://www.bioinformatics.babraham.ac.uk/projects/fastqc/) to evaluate read quality and preprocessed using fastp [31] for quality filtering, adapter trimming, and removal of low-quality read bases. Filtered reads were aligned to mm10 using Minimap2 [32]. The resulting SAM files were converted to BAM and sorted using Samtools [33] to facilitate efficient downstream processing. Transcript isoform identification and quantification were performed using the Bambu R package [34], which enables accurate annotation and expression of novel and known transcripts from long read sequencing data. Visualization of the data was performed using ggplot2 R package (https://www.rdocumentation.org/packages/ggplot2/versions/3.5.0).

## Supporting information

**S1 Table. Long read sequencing detection of *Pvt1* isoforms.** Annotated *Pvt1* reads, indicating Ensemble transcript annotations and number of reads in *Kras$^{LA2/+}$; p53$^{LSL/LSL}$; Rosa26-CreER$^{T2}$* (KPR) lung adenocarcinoma (LUAD) cell line in the absence and presence of Tamoxifen (Tam)-mediated p53 restoration and activation by oncogenic stress. Bolded lines highlight the 18 transcripts with ≥5 reads, plotted in Fig 6C and 6D.
(XLSX)

**S2 Table. Oligonucleotides.** List of constructs, gRNAs, primers, and smRNA-FISH probes in this study.
(XLSX)

**S1 Fig. CRISPR/Cas9 screen to identify *Pvt1b*-specific functional elements in KPR cells.**
(PDF)

## Acknowledgments

We thank Justin Glynn and other members of the Dimitrova lab for insightful comments. We are grateful to Rick Maser from Jackson Laboratories for the generation of *Pvt1b* PAS mutant mice, the Yale Center for Genome Analysis (YCGA), and the Yale Pathology Tissue Services.

## Author contributions

**Conceptualization:** Qiao Li, Christiane E. Olivero, Nadya Dimitrova.

**Data curation:** Qiao Li, Nadya Dimitrova.

**Formal analysis:** Qiao Li, Christiane E. Olivero, Erin Floyd, Jane Ding, Emily Dangelmaier, James Knight, Nadya Dimitrova.

**Funding acquisition:** Nadya Dimitrova.

**Investigation:** Qiao Li, Christiane E. Olivero, Jane Ding, Emily Dangelmaier, Nadya Dimitrova.

**Methodology:** Qiao Li, Christiane E. Olivero, Nadya Dimitrova.

**Project administration:** Nadya Dimitrova.

**Resources:** Nadya Dimitrova.

**Supervision:** Nadya Dimitrova.

**Validation:** Nadya Dimitrova.

**Visualization:** Nadya Dimitrova.

**Writing – original draft:** Nadya Dimitrova.

**Writing – review & editing:** Nadya Dimitrova.

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
