## [Decision Letter · Decision Letter 0]

PGENETICS-D-25-00270

Activation of Pvt1b isoform contributes to local Pvt1 abundance to repress Myc during stress

PLOS Genetics

Dear Dr. Dimitrova,

Thank you for submitting your manuscript to PLOS Genetics. After careful consideration, we feel that it has merit but does not fully meet PLOS Genetics's publication criteria as it currently stands. Therefore, we invite you to submit a revised version of the manuscript that addresses the points raised during the review process.

Please submit your revised manuscript within 30 days May 30 2025 11:59PM. If you will need more time than this to complete your revisions, please reply to this message or contact the journal office at plosgenetics@plos.org. Please include the following items when submitting your revised manuscript:

We look forward to receiving your revised manuscript.

Kind regards,

Yan Tang

Academic Editor

PLOS Genetics

Monica Colaiácovo

Section Editor

PLOS Genetics

Aimée Dudley

Editor-in-Chief

PLOS Genetics

Anne Goriely

Editor-in-Chief

PLOS Genetics

**Journal Requirements:**

At this stage, the following Authors/Authors require contributions: Nadya Dimitrova. Please ensure that the full contributions of each author are acknowledged in the "Add/Edit/Remove Authors" section of our submission form.

The list of CRediT author contributions may be found here: https://journals.plos.org/plosgenetics/s/authorship#loc-author-contributions

https://journals.plos.org/plosgenetics/s/submission-guidelines#loc-parts-of-a-submission

5) We have noticed that you have uploaded Supporting Information files, but you have not included a list of legends. Please add a full list of legends for your Supporting Information files after the references list.

6) We notice that your supplementary Figure is included in the manuscript file. Please remove it and upload it with the file type 'Supporting Information'. Please ensure that each Supporting Information file has a legend listed in the manuscript after the references list. 

7) Some material included in your submission may be copyrighted. According to PLOSu2019s copyright policy, authors who use figures or other material (e.g., graphics, clipart, maps) from another author or copyright holder must demonstrate or obtain permission to publish this material under the Creative Commons Attribution 4.0 International (CC BY 4.0) License used by PLOS journals. Please closely review the details of PLOSu2019s copyright requirements here: PLOS Licenses and Copyright. If you need to request permissions from a copyright holder, you may use PLOS's Copyright Content Permission form.

Potential Copyright Issues:

i) Figure 5A. Please confirm whether you drew the images / clip-art within the figure panels by hand. If you did not draw the images, please provide (a) a link to the source of the images or icons and their license / terms of use; or (b) written permission from the copyright holder to publish the images or icons under our CC BY 4.0 license. Alternatively, you may replace the images with open source alternatives. See these open source resources you may use to replace images / clip-art:

8) Please amend your detailed Financial Disclosure statement. This is published with the article. It must therefore be completed in full sentences and contain the exact wording you wish to be published.

1) State what role the funders took in the study. If the funders had no role in your study, please state: "The funders had no role in study design, data collection and analysis, decision to publish, or preparation of the manuscript.".

**Reviewers' comments:**

Reviewer's Responses to Questions

Reviewer #1: This study investigates the role of the Plasmacytoma variant translocation 1 (PVT1) lncRNA, specifically the Ptv1b isoform, in the p53-dependent repression of Myc expression following stress induction. Based upon the author’s prior work demonstrating p53 targeting of Ptv1b under stress, they explore the mechanism of Myc transcriptional repression by Ptv1b, focusing on whether this effect is mediated by repressive sequence elements within exon 1b or by an increase in local Ptv1 abundance.

The authors conducted a comprehensive analysis employing well-designed in vitro and in vivo experiments with robust methodology and appropriate controls, which is a strength of the study. The use of multiple genetic mouse models further supports the findings. While acknowledging potential limitations and carefully interpreting some inconsistent data, the authors provide thorough validation of their observations across numerous experiments. The manuscript is logically structured, clearly presented, and the conclusions are well supported by the evidence data.

Below are some minor comments:

1. PVT1b isoform expression and regulation: Literature data demonstrate significant differential expression patterns of PVT1 isoforms across various human tissues. To provide a clearer context for the study's focus on Ptv1b, the authors could clarify the general expression differences between Ptv1a and Ptv1b. It would be valuable for the authors to discuss existing studies that indicate tissue-specific expression patterns for the Ptv1b isoform specifically. What differences were noted in the expression and abundance of Ptv1a and Ptv1b in this study? Is Ptv1a generally expressed at higher levels than Ptv1b, even following stress induction? Or were the read counts for Ptv1b greater than those for Ptv1a when stress was induced? Furthermore, does Pvt1b have any known role in other regulatory mechanisms beyond controlling Myc expression at the transcriptional level? This information might interest the reader and provide a broader context for Ptv1b function and its regulation.

2. p53 binding to Ptv1b promoter: The authors' findings that p53 controls the expression of Ptv1b isoform under stress could be strengthened by citing any publicly accessible ChIP-seq data for p53 that corroborates these findings at the genomic level. This would provide an additional independent layer of evidence for their research conclusions.

3. Myc expression in Pvt1b-deficient mice: Do authors expect differences in Myc expression/nuclear localization in tumors from LA1 Pvt1b+/+ and LA1 Pvt1bP+/- mice? This could be potentially tested by IHC staining.

4. PVT1 as an oncogene and potential tumor suppressor roles: PVT1 in human is located on chromosome 8q24, a region frequently amplified in various cancers, often alongside the MYC oncogene. Therefore, PVT1 is predominantly recognized as an oncogene across a wide range of human cancers. However, authors present here evidence for tumor suppressor-like activities for Pvt1. Given the complex and context-dependent nature of lncRNA function, the authors should elaborate on the specific circumstances or cancer types where PVT1 (or potentially specific isoforms or its promoter) has been reported to exhibit tumor suppressor-like activities. This nuanced discussion would provide a more complete picture of PVT1's multifaceted role in cancer.

In conclusion, the comments mentioned above are minor and do not significantly detract from the overall quality and impact of this study. I recommend this manuscript for publication.

Reviewer #2: Figure 1: Add quantitative data showing the relative expression levels of Myc across all mutant lines in one comparative graph.

Figure 2: Add a direct side-by-side quantitative comparison between Pvt1a and Pvt1b repressive activity; Include representative images of the smRNA-FISH alongside the quantification.

Figure 4: Create a panel showing protein levels of Myc alongside RNA measurements; Enhance the senescence data with additional markers beyond β-galactosidase.

Figure 5: Add immunohistochemistry for Myc in tumor sections to directly link the phenotype to Myc regulation; Include survival data for both genotypes to assess functional significance.

Consider adding a graphical abstract summarizing the overall findings.

Reviewer #3: Li et al. present a genetic analysis of the Pvt1 locus and its relation to Myc repression upon genotoxic stress. The work builds upon a body of prior data from the Dimitrova lab which previously showed that a spliced isoform from the Pvt1 lncRNA gene, Pvt1b, repressed Myc in a p53-dependent manner after stress. Here, Li and colleagues confirm results from those prior studies, and go on to demonstrate in a mouse model that the Pvt1b isoform plays a tumor-suppressive role in lung adenocarcinoma in vivo. At the same time, exhaustive mutational analyses demonstrate that the Pvt1b isoform per se is not required for repression of Myc, and suggest that alternate Pvt1 isoforms or overall transcription from the Pvt1 locus mediate Myc repression in a sequence-nonspecific manner. The authors interpret these data to support a model first put forth by Rick Young and colleagues, in which high local RNA levels can disrupt transcriptional condensates of a nearby promoter, in this case, Myc. The study was clearly written, elegant, and rigorously controlled. The conclusions were well supported by the data and the findings are of high significance to the field.

**Have all data underlying the figures and results presented in the manuscript been provided?**

Reviewer #1: Yes

Reviewer #2: Yes

Reviewer #3: Yes

PLOS authors have the option to publish the peer review history of their article (what does this mean? ). If published, this will include your full peer review and any attached files.

**Do you want your identity to be public for this peer review?** For information about this choice, including consent withdrawal, please see our Privacy Policy .

Reviewer #1: No

Reviewer #2: No

Reviewer #3: No

**Figure resubmission:**
---

## [Editor Report · Decision Letter 1]

Dear Dr Dimitrova,

We are pleased to inform you that your manuscript entitled "Activation of Pvt1b isoform contributes to local Pvt1 abundance to repress Myc during stress" has been editorially accepted for publication in PLOS Genetics. Congratulations!

Yours sincerely,

Monica Colaiácovo

Section Editor

PLOS Genetics

Aimée Dudley

Editor-in-Chief

PLOS Genetics

Anne Goriely

Editor-in-Chief

PLOS Genetics

Comments from the reviewers (if applicable):

**Data Deposition**

http://datadryad.org/submit?journalID=pgenetics&manu=PGENETICS-D-25-00270R1

**Press Queries**

---

## [Editor Report · Acceptance letter]

PGENETICS-D-25-00270R1

Activation of Pvt1b isoform contributes to local Pvt1 abundance to repress Myc during stress

Dear Dr Dimitrova,

We are pleased to inform you that your manuscript entitled "Activation of Pvt1b isoform contributes to local Pvt1 abundance to repress Myc during stress" has been formally accepted for publication in PLOS Genetics! Your manuscript is now with our production department and you will be notified of the publication date in due course.

With kind regards,

Anita Estes

PLOS Genetics

On behalf of:
